# Evaluation of Anti-S1 IgA Response to Different COVID-19 Vaccination Regimens

**DOI:** 10.3390/vaccines11061117

**Published:** 2023-06-19

**Authors:** Teeraporn C. Bureerug, Sitthichai Kanokudom, Nungruthai Suntronwong, Ritthideach Yorsaeng, Suvichada Assawakosri, Thanunrat Thongmee, Yong Poovorawan

**Affiliations:** 1Department of Microbiology, Faculty of Medicine, Srinakharinwirot University, Bangkok 10110, Thailand; 2Center of Excellence in Clinical Virology, Department of Pediatrics, Faculty of Medicine, Chulalongkorn University, Bangkok 10330, Thailand; kanokudom_s@yahoo.com (S.K.); suntronwong.n@gmail.com (N.S.); ritthideach.yor@gmail.com (R.Y.); suvichada.assawa@gmail.com (S.A.); tata033@hotmail.com (T.T.); yong.p@chula.ac.th (Y.P.); 3Center of Excellence in Osteoarthritis and Musculoskeleton, Faculty of Medicine, Chulalongkorn University, King Chulalongkorn Memorial Hospital, Thai Red Cross Society, Bangkok 10330, Thailand; 4Fellow of the Royal Society of Thailand, The Royal Society of Thailand, Sanam Sueapa, Dusit, Bangkok 1030, Thailand

**Keywords:** severe acute respiratory virus 2 (SARS-CoV-2), COVID-19 vaccine, spike 1 specific immunoglobulin A (anti-S1 IgA), homologous, heterologous, intramuscular (IM), intradermal (ID)

## Abstract

IgA plays a crucial role in early virus neutralization. To identify the IgA stimulation by COVID-19 vaccine, this study aimed to evaluate the level of anti-S1 IgA in the serum of participants immunized with different COVID-19 vaccination regimens. Sera from 567 eligible participants vaccinated with two, three, or four doses of different types of COVID-19 vaccine were recruited. Post-vaccine anti-S1 IgA responses significantly varied according to vaccine type and regimen. The finding showed that heterologous boosters, especially after priming with an inactivated vaccine, elicited higher IgA levels than homologous boosters. Vaccination with SV/SV/PF produced the highest IgA level among all the immunization regimens after either two, three, or four doses. The different routes and amounts of vaccine used for vaccination showed non-significant differences in IgA levels. After the third dose of immunization for 4 months, the level of IgA decreased significantly from the level found on day 28 in both SV/SV/AZ and SV/SV/PF groups. In conclusion, our study showed that heterologous booster regimens for COVID-19 elicited higher anti-S1 IgA levels in serum, especially after priming with inactivated vaccine. The presented anti-S1 IgA may have advantages in preventing SARS-CoV-2 infection and severe disease.

## 1. Introduction

Close to the end of 2019, the Coronavirus disease 2019 (COVID-19) pandemic caused significant morbidity and mortality worldwide and major social, educational, and economic disruptions [1]. In this context, vaccine rollout represents a tool of choice to prevent infection and control the pandemic. Approval of these vaccines offers a highly effective tool for the global control of this event [2]. Several vaccine candidates have been approved for emergency use and are currently being administered to populations worldwide. Thailand launched its vaccination program with many vaccines in March 2021. Until now, many types of vaccines have been approved by the Thai Food and Drug Administration (FDA); they include inactivated vaccines (CoronaVac (Sinovac Life Sciences Co., Ltd., Beijing, China) and BBIBP-CorV (Beijing Institute of Biological Products Co., Ltd. (Sinopharm), Beijing, China)), the viral-vector vaccines (AZD1222 (AstraZeneca, University of Oxford, Oxford, UK) and Ad26.CoV2.S (Johnson & Johnson, New Brunswick, NJ, USA)), mRNA vaccines (BNT162b2 (Pfizer-BioNTech Inc., New York, NY, USA) and mRNA−1273 (Moderna Inc., Cambridge, MA, USA)), and the protein subunit vaccine Covovax^TM^ (Serum Institute of India Pvt Ltd., Pune, India) [3]. The heterologous vaccine used in Thailand shows a good immune response [4].

IgA is present in both serum and mucosal secretions. It is the second most prevalent antibody found in circulation, following IgG, and the predominant immunoglobulin in secretion [5]. It plays an essential protective role against bacteria and viruses [6,7,8] that target mucosal tissues, such as influenza [9]. Secretory IgA is critical in protecting mucosal surfaces by neutralizing viruses or preventing viral attachment to the mucosal epithelium [10]. Virus-targeting neutralizing antibodies have been established to be essential in clearance and recovery after viral infection. Ainai et al. [11] suggested that an elevated IgA serum level correlates with influenza vaccine efficacy, which is assumed to be the same as that of the COVID-19 vaccine. Sterlin et al. [12] showed that SARS-CoV-2 neutralization has a better correlation with IgA than IgM or IgG in the first weeks after symptom onset, because human IgA antibodies are often detectable before the appearance of SARS-CoV-2-specific IgG—suggesting a role for IgA antibodies in early virus neutralization. The developing mucosal immunity via IgA may be important in preventing COVID-19 infection. Consequently, elevated IgA level after immunization with mRNA vaccines in both systemic and mucosal surfaces, such as the respiratory tract, plays a defensive role against SARS-CoV-2 [13,14]. In this study, we aimed to evaluate the level of anti-S1 IgA in the serum of participants immunized with two, three, or four doses of a combination of COVID-19 vaccines. Additionally, we determined the IgA response from different routes to find the best vaccine regimens that induced the highest IgA level for basic knowledge and practical use in the future.

## 2. Materials and Methods

### 2.1. Study Design

Our study aimed to evaluate the SARS-CoV-2 Spike 1-specific immunoglobulin A (anti-S1 IgA) by using the leftover serum from our previous cohort studies on the safety and immunogenicity of COVID-19 vaccines [15,16,17,18,19,20]. In our previous studies, we reported the levels of binding antibodies, such as IgG, and neutralizing antibodies, following immunization with different regimens of two, three, and four doses of COVID-19 vaccines. The study was conducted at the Center of Excellence in Clinical Virology, Faculty of Medicine, Chulalongkorn University in Bangkok, Thailand between March 2021 and September 2022 [15,16,17,18,19,20]. The participants were conveniently assigned to receive any vaccine that was available during the study period. The research protocols were approved by the Research Ethics Committee of the Faculty of Medicine, Chulalongkorn University (IRB no. 572/63, 192/64, 491/64, 546/64, and 223/65). All participants provided written informed consent.

### 2.2. Participants and Types of Vaccine

To evaluate the anti-S1 IgA response after different vaccine schedules, the study regimens comprised the primary and booster (third and fourth dose) regimens. The primary series comprised two doses of CoronaVac (hereafter referred to as SV/SV), two doses of AZD1222 (hereafter referred to as AZ/AZ), heterologous CoronaVac followed by AZD1222 (hereafter referred to as SV/AZ), heterologous AZD1222 followed by CoronaVac (hereafter referred to as AZ/SV), heterologous CoronaVac followed by BNT162b2 (hereafter referred to as SV/PF), and two doses of BNT162b2 (hereafter referred to as PF/PF). Additionally, the booster groups comprised BBIBP-CorV after two doses of primed CoronaVac (hereafter referred to as SV/SV/SP), AZD1222 after two doses of primed CoronaVac (hereafter referred to as SV/SV/AZ), BNT162b2 after two doses of primed CoronaVac (hereafter referred to as SV/SV/PF), four doses of AZD1222/AZD1222/BNT162b2/BNT162b2 (hereafter referred to as AZ/AZ/PF/PF), and four doses of AZD1222/AZD1222/mRNA−1273/50 μg mRNA−1273 (hereafter referred to as AZ/AZ/MO/MO½).

All individuals were administered with vaccines intramuscularly, following the vaccination protocol. The sera collected from unvaccinated individuals (hereafter referred to as Naïve) and convalescent sera were used as negative and positive controls, respectively. The inclusion criteria were healthy adults (≥18 years) and no history of COVID-19. All included sera were collected at 28 ± 7 days after the last dose. Sera from individuals who did not provide complete demographic and clinical data were excluded. Additionally, the participants who tested seropositive for anti-nucleocapsid antibodies targeting SARS-CoV-2, which were used to confirm the infection, were excluded from the study. In parallel, different routes of vaccination were evaluated. Intradermal (ID) administration with AZD1222 and BNT162b2 after two doses of primed CoronaVac (hereafter referred to as SV/SV/AZ(ID) and SV/SV/PF(ID), respectively) was compared to standard intramuscular administration. To observe the waning of vaccine-induced anti-S1 IgA levels, the individuals from the SV/SV/SP, SV/SV/AZ, and SV/SV/PF groups were followed up for 90–120 days (D90 and D120).

### 2.3. The SARS-CoV-2 Anti-S1 IgA Assay

Sera collected from vaccinated individuals were analyzed to determine the level of anti-S1 IgA. The anti-S1 IgA was assessed using the enzyme-linked immunosorbent assay (ELISA) (Euroimmun, Lübeck, Germany), following the kit’s instructions. The seropositivity of anti-S1 IgA was considered as ≥1.1 S/C. After reaching the upper limit ratio (S/C), the samples were diluted (dilution factor equal to 4) and re-monitored. Therefore, the current study’s final upper limit ratio (S/C) was reported as 28.

### 2.4. Statistical Analysis

Baseline characteristics were reported as means with standard deviations (SD) or medians with interquartile ranges (IQR). A categorical analysis of sex and age was performed using Pearson’s Chi-square test and the Kruskal–Wallis test, followed by Dunn’s post hoc test with Bonferroni’s correction. Between-group differences in the median of anti-S1 IgA (S/C) response at 28 days after vaccination among the two, three, and four-dose regimens were calculated using the Kruskal–Wallis test, followed by Dunn’s post hoc test with Bonferroni’s correction. The median anti-S1 IgA levels measured on days 90–120 and day 28 were statistically evaluated using Mann–Whitney U-test. A *p*-value of <0.05 was considered statistically significant.

## 3. Results

### 3.1. Participant Characteristics

A total of 567 eligible sera were obtained for the anti-S1 IgA investigation. The participant flow diagram of this study was presented in Figure 1. The COVID-19 vaccination guidelines for two, three, and four doses of these participants are described in Appendix A. The demographic and clinical data of Naïve and vaccinated individuals who were immunized with two, three, and four doses are shown in Table 1, Table 2 and Table 3.

Among the individuals vaccinated with two doses, the number of women in the SV/PF group was significantly higher (95.0%) than that of women in other groups (45.2–59.3%), except for the PF/PF group (63.2%) (Table 1). The number of women was similar among all groups, except for the SV/PF group. The mean age of the SV/PF group (24.2 years) was significantly lower than that of the other six groups (*p* < 0.05). Additionally, the mean age of the PF/PF group (34.4 years) was significantly lower than that of the AZ/AZ group (46.9 years, *p* < 0.001) but not different from those of the other five groups (41.5–45.5 years). In individuals who received one and two booster doses of the COVID-19 vaccine, no significant differences in sex and age were observed among groups (Table 2 and Table 3). The median interval between the last dose and blood collection ranged between 28.0 days to 34.0 days (Table 1, Table 2 and Table 3). In order to investigate the durability of the anti-S1 IgA, a subgroup of participants (*n* = 23 per group) who received the third dose using SP, AZ, and PF vaccines in the SV/SV regimen were followed up at day 90–120. The characteristics of these subgroups are presented in Appendix A.

### 3.2. Robust Anti-S1 IgA Production after a Second Dose of Vaccine Combination

When comparing anti-S1 IgA in the sera of participants who received two doses of vaccine, we found that a combination of inactivated vaccine (SV) with a second dose of mRNA vaccine (PF) induced the highest median level of IgA (Figure 2). Vaccination with PF/PF stimulated a significantly higher level of IgA than vaccination with SV/SV and AZ/AZ. Vaccination with SV/PF stimulated a significantly higher level of IgA than vaccination with SV/SV, SV/AZ, and AZ/AZ. Similarly, SV/AZ stimulated significantly higher levels of IgA than SV/SV and AZ/AZ. The results showed that heterologous boosters elicited higher IgA levels than homologous boosters, especially after priming with an inactivated vaccine. Interestingly, vaccination with SV/AZ stimulated significantly higher IgA levels than vaccination with AZ/SV.

### 3.3. Higher Expression of Anti-S1 IgA after mRNA Booster Regimens

When comparing anti-S1 IgA induced by all immunization regimens, we found that the regimen that included the mRNA-type vaccine (PF) stimulated higher IgA levels, as shown in Figure 3. Furthermore, vaccination with SV/SV/PF stimulated a higher median IgA level than all other immunization regimens, after either including two, three, or four doses of vaccination. Between both groups, the IgA levels in the sera of participants vaccinated with four doses of the vaccine (AZ/AZ/PF/PF and AZ/AZ/MO/MO½) with different amounts of mRNA (PF = 30 μg, half dose of MO = 50 μg) were not significantly different. Additionally, the IgA level of the four dose groups were not statistically different from those found in convalescent serum.

### 3.4. Different Routes and Amounts of Vaccine Produced Similar Results

To examine whether decreased amounts of vaccine affected the immune response, especially the amount of anti-S1 IgA, we compared the intramuscular (IM) ID routes of the third dose of both AZ and PF. Different routes of vaccination and amounts of vaccine were used. The dose of viral-vector vaccine, AZ, was 0.5 mL for IM and 0.1 mL for ID. The dose of mRNA vaccine, PF, was 30 μg (0.3 mL) for IM and 10 μg (0.1 mL) for ID. Therefore, the median anti-S1 IgA level was not significantly different between both groups (IM and ID) after vaccination with AZ and PF, as shown in Figure 4. Therefore, the upper limit ratio of IgA level after booster administration was achieved by more participants in the IM route group than those in the ID route group.

### 3.5. Waning of Anti-S1 IgA Level in the Sera of Participants with Three Doses of Immunization

We determined how long the antibody persisted in the sera of participants, as shown in Figure 5. After the third dose of immunization for 4 months, the level of IgA decreased significantly from the level found on day 28 in both the SV/SV/AZ and SV/SV/PF groups. However, the anti-S1 IgA level was still more than the level found in the Naïve group. IgA persisted in the sera of one participant in both SV/SV/AZ and SV/SV/PF groups until after 4 months. The SV/SV/SP regimen showed the lowest level of IgA, which decreased at day 90. However, the level was not significantly different from the level determined on day 28.

## 4. Discussion

In this study, we measured the levels of anti-S1 IgA in participants immunized with two, three, or four doses of the homologous and heterologous COVID-19 vaccines. Additionally, we observed the IgA response induced by different routes of immunization and the persistence of IgA immunity. We were interested in IgA antibodies because mucosal surfaces are key participants in SARS-CoV-2 infection. Therefore, a host mucosal immune defense could be protective. Additionally, Sterlin et al. [12] showed an early humoral response to SARS-CoV-2 dominated by IgA-expressing plasmablasts. Therefore, IgA antibodies are often detectable before the appearance of SARS-CoV-2-specific IgG, suggesting that IgA contributes more to serum neutralization potential than IgG in the early phase of the infection. Therefore, we measured IgA in the sera instead of in secretion because our previous study showed that paired bio-sampling of saliva and blood in patients with COVID-19 had strong positive correlations with SARS-CoV-2 Spike (S) and receptor-binding domain (RBD)-specific immunoglobulin (IgM, IgG, and IgA) levels [21].

Based on 567 eligible sera from vaccinated participants, we showed that post-vaccine anti-S1 IgA responses significantly vary according to vaccine type and regimen. No significant differences in sex and age were observed among groups of vaccine regimens. Our data further confirmed that the mix-and-match COVID-19 vaccination strategy triggered a stronger antibody production than two doses of a single vaccine. Interestingly, all regimens primed with the inactivated vaccine (SV) and boosted with the mRNA vaccine (PF or MO) resulted in a higher median level of anti-S1 IgA than the others. In addition to IgA response, our previous reports have reported that individuals who received inactivated vaccine, followed by a booster of mRNA vaccine, could exhibit high levels of binding antibodies, such as IgG, IgM, and total Ig, and possess broad neutralizing potency [3,17,18,19]. Therefore, the sequence in giving the primary and booster doses with different types of vaccine was significant, as it was observed in this study that SV/AZ induced higher IgA levels than AZ/SV. Priming with inactivated viral particle (SV) may induce the immune system to recognize all antigenic determinants of the virus. As a result, many memory cells can produce a robust immune response after boosting with spike proteins from the viral-vector vaccine (AZ) or mRNA vaccine (PF).

The anti-S1 IgA levels from natural infection (convalescent) were not significantly different between the primary vaccine and mRNA booster vaccine (PF/PF). Moreover, the levels were significantly lower than those observed after the priming dose with inactivated vaccine and one or two booster doses of mRNA vaccine. Our study demonstrated that three vaccination doses with SV/SV/PF induced the highest level of anti-S1 IgA. Additionally, we compared the IM and ID routes of vaccine administration for the third booster dose after priming with two doses of inactivated vaccine. Both the viral-vector (AZ) and mRNA vaccines (PF) were used. We found that the routes of vaccine immunization for either viral-vector or mRNA vaccines were not significantly different in the level of anti-S1 IgA. However, the IM groups with the PF vaccine demonstrated a slightly higher level of anti-S1 IgA when compared to the ID route (a median fold increase of 1.5). These findings align with the previous studies conducted by Intapiboon et al. and Assantachai et al. [22,23], which demonstrated that IM administration resulted in significantly higher anti-RBD IgG level compared to the ID route. This ID-boosting strategy provides a suitable alternative for vaccines and effective vaccine management to increase coverage during vaccine shortage. However, ID injection can be challenging to perform and requires skilled medical professionals.

Two vaccine regimens, AZ/AZ/MO/MO½ and AZ/AZ/PF/PF, were administered in four doses. The levels of anti-S1 IgA in both groups were not significantly different. However, the amount of mRNA vaccine in the half dose of MO was higher than that of PF.

Antibody levels after COVID-19 vaccination may drop at different rates depending on various factors, including the type of vaccine, infection before or after vaccination, age, sex, T-cell response, and the interval between vaccine administration [24,25]. In this study, we found that, on day 120 post-vaccination with viral-vector or mRNA vaccines, anti-S1 IgA level reduced significantly from day 28 post-vaccination. However, some participants still had anti-S1 IgA levels that were higher than the upper measurement limit. Additionally, this study demonstrated that in a group that received three doses of inactivated vaccine (SV/SV/SP), anti-S1 IgA level was low at 28 days post-vaccination. Therefore, additional investigations are required to better define the stability of immune effectors after COVID-19 vaccination, because other arms of specific immune response, T-cell reactivity, and T-cell memory may represent other important mechanisms for long-lasting vaccine-induced protection.

Because sIgA plays a more important role in mucosal immunity, especially for viral neutralization, a suitable vaccine should generate IgA-plasmablasts with many homing receptors to the mucosal surface to prevent person-to-person transmission, as in natural infection. Currently, many researchers are interested in mucosal vaccines, such as intranasal (IN) immunization [26,27]. Sheikh-Mohamed et al. [28] suggested that an IM primer followed by an IN booster strategy is worth considering for preventing person-to-person transmission of SARS-CoV−2 and broad protection against emerging variants. This strategy is supported by a previous study that showed that antigens previously introduced by the mucosal route can prime the immune system so that subsequent systemic immunization induces both systemic and mucosal antibody responses [29,30]. From our data, mRNA vaccines administered by the IM route stimulated higher IgA levels than the other types of vaccine. Priming with an IN vaccine to generate IgA plasmablasts and boosting with an IM mRNA vaccine has potential in improving immunity against SARS-CoV-2.

This study had some limitations. First, homologous boosters, such as SV/SV/SV, AZ/AZ/AZ, and PF/PF/PF, were not available for comparison of IgA levels with heterologous boosters. Second, we did not collect saliva samples to assess mucosal immunity. Third, we only measured the amount of IgA but did not evaluate its neutralizing and/or functional activities.

## 5. Conclusions

Our findings showed that post-vaccine anti-S1 IgA responses significantly varied according to vaccine type and vaccine regimen. The heterologous booster regimens for COVID-19 elicited the highest anti-S1 IgA level, particularly priming with inactivated vaccine. Additionally, the mRNA vaccine (PF) stimulated a higher amount of IgA than other vaccine types (inactivated and viral-vector vaccines). However, the different routes and amounts of the booster vaccine did not significantly affect IgA levels. The booster doses are required to increase IgA levels and prolong immunity, which might possibly help to decrease disease severity. Antibodies to vaccines may persist in sera for more than 120 days. These findings suggest that a heterologous booster with an inactivated vaccine, followed by an mRNA vaccine, may lead to robust anti-S1 IgA production, and that the mRNA vaccine may be a more effective booster.

## Figures and Tables

**Figure 1 vaccines-11-01117-f001:**
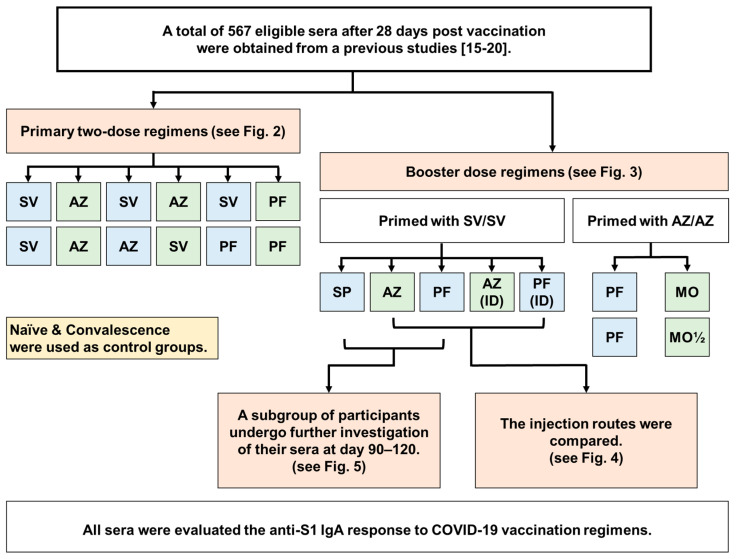
Participant flow diagram of the study. We obtained a total of 567 leftover serum samples from our previous studies [15,16,17,18,19,20] for the purpose to evaluating of anti-S1 IgA response at 28 days after being immunized with two, three, or four doses of COVID-19 vaccine. These samples were collected at 28 ± 7 days after being immunized with two, three, or four doses of COVID-19 vaccine. The administration routes between intramuscular (IM) and intradermal (ID) after a booster dose with AZ and PF were compared. Additionally, we assessed the waning immunity at 90–120 days after the third dose using SP, AZ, and PF vaccines in the SV/SV regimen. Abbreviations: AZ: AZD1222; PF: BNT162b2; SP: BBIBP-CorV; SV: CoronaVac; MO: mRNA−1273; MO½: 50 µg mRNA−1273.

**Figure 2 vaccines-11-01117-f002:**
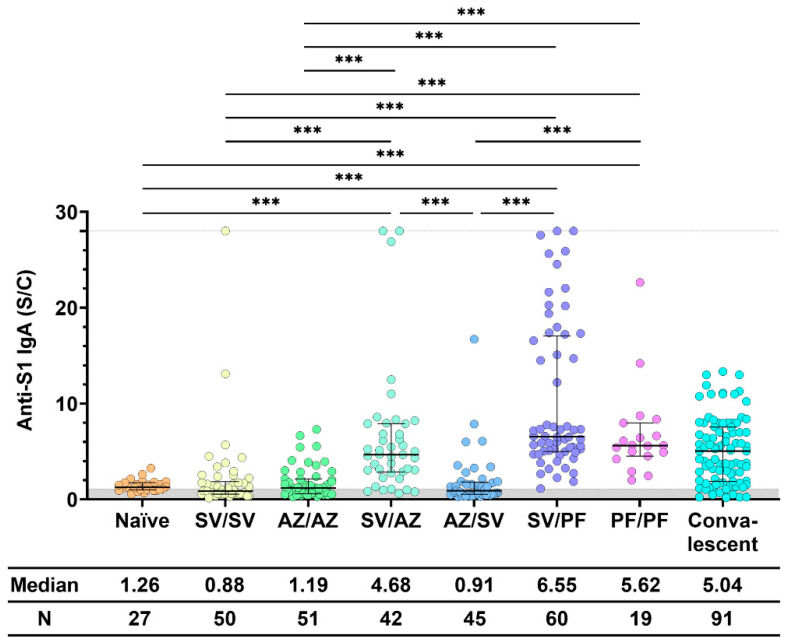
Anti-S1 IgA response after two doses of COVID-19 vaccine. Sera were collected before (Naïve) and after 28 ± 7 days of COVID-19 vaccination, while convalescent sera were collected at 5–8 weeks after being exposed to COVID-19. All individuals received the standard dose following the COVID-19 vaccine instruction. The gray area indicates the seronegativity of anti-S1 IgA (<1.1 S/C). The lines represent the median (interquartile range [IQR]). Significant differences between groups in anti-S1 IgA were calculated by the Kruskal–Wallis test, followed by Dunn’s post hoc test with Bonferroni’s correction. Pairwise comparisons display significant values, including *p* < 0.001 (***). CoronaVac (SV), AZD1222 (AZ), and BNT162b2 (PF).

**Figure 3 vaccines-11-01117-f003:**
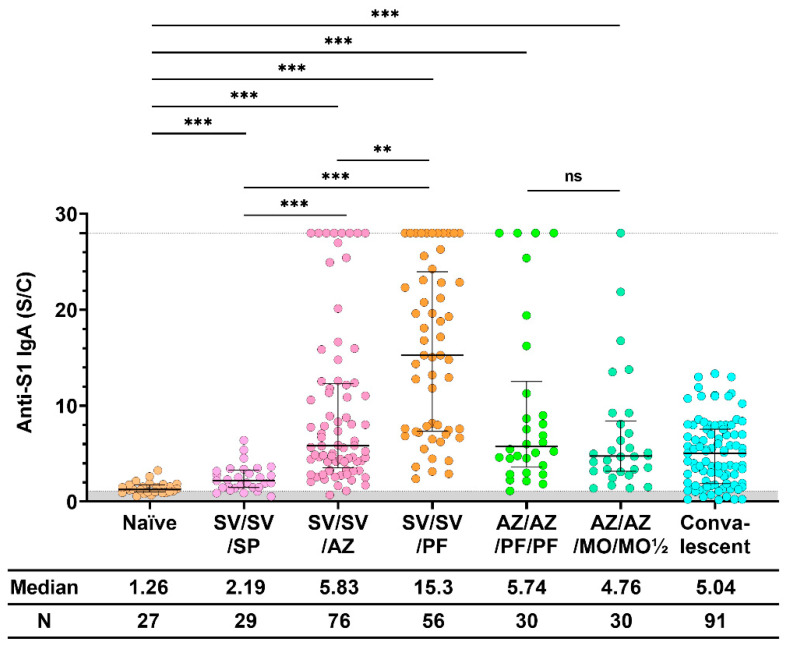
Anti-S1 IgA response after the booster dose. Sera were collected before (Naïve) and after 28 ± 7 days of COVID-19 vaccination, while convalescent sera were collected at 5–8 weeks after being exposed to COVID-19. All individuals received the standard dose following the COVID-19 vaccine instruction. The gray area indicates the seronegativity of anti-S1 IgA (<1.1 S/C). The lines represent the median (interquartile range [IQR]). Significant differences between groups in anti-S1 IgA were calculated by the Kruskal–Wallis test, followed by Dunn’s post hoc test with Bonferroni’s correction. Pairwise comparisons display significant values, including *p* < 0.01 (**), *p* < 0.001 (***), and no statistical significance (ns). CoronaVac (SV), AZD1222 (AZ), BNT162b2 (PF), and mRNA−1273 (MO).

**Figure 4 vaccines-11-01117-f004:**
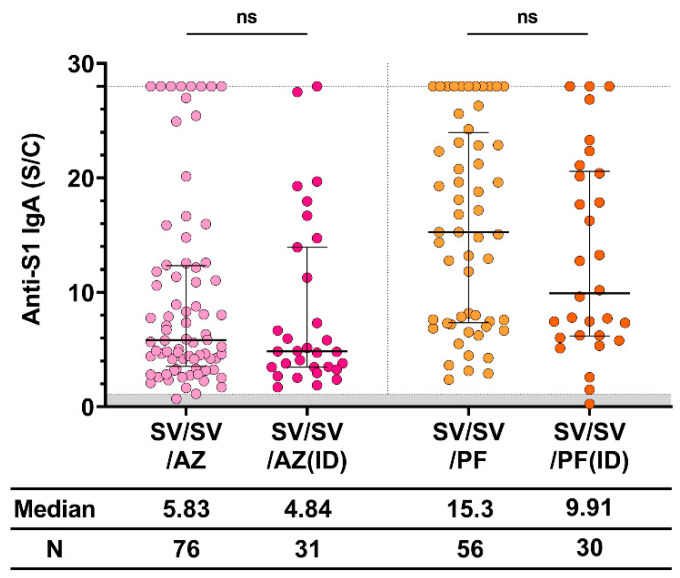
Anti-S1 IgA response after IM and ID vaccination. The standard dose of AZD1222 and BNT162b2 was intramuscularly administered, whereas the reduced doses (0.1 mL [10^10^ IU] for AZD1222 and 0.1 mL [10 μg] for BNT162b2) were ID administered. The gray area indicates the seronegativity of anti-S1 IgA (<1.1 S/C). The lines represent the median (interquartile range [IQR]). Significant differences between groups were calculated by non-parametric Mann–Whitney U-test. Pairwise comparisons display no statistical significance (ns). CoronaVac (SV), AZD1222 (AZ), and BNT162b2 (PF).

**Figure 5 vaccines-11-01117-f005:**
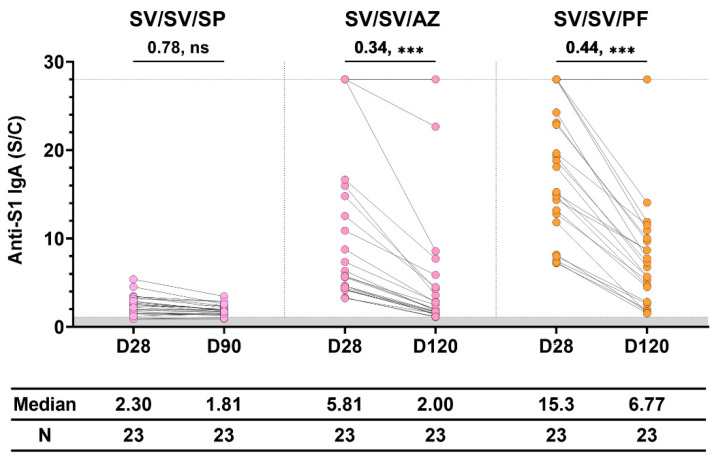
The waning of vaccine-induced anti-S1 IgA in individuals immunized with different three-dose vaccination. Sera were collected at 28 ± 7 days (D28) and 90–120 days (D90 and D120) after vaccination. The gray area indicates the seronegativity of anti-S1 IgA (<1.1 S/C). The lines represent the median (interquartile range [IQR]). Significant differences between groups were calculated by non-parametric Mann–Whitney U-test. Pairwise comparisons display the median fold change and significant values, including *p* < 0.001 (***) and no statistical significance (ns).

**Table 1 vaccines-11-01117-t001:** Characteristics of the participants who received two doses of COVID-19 vaccine.

Characteristics	Naïve	SV/SV	AZ/AZ	SV/AZ	AZ/SV	SV/PF	PF/PF
Number	27	50	51	42	45	60	19
Sex, Female no. (%)	16 (59.3)	23 (46.0)	30 (58.8)	19 (45.2)	26 (57.8)	57 (95.0)	12 (63.2)
Age years, mean	45.5	42.1	46.9	41.5	43.5	24.2	34.4
[SD]	[12.0]	[9.1]	[16.4]	[12.5]	[8.9]	[6.3]	[11.9]
(min–max)	(30.0–68.0)	(24.0–59.0)	(22.0–85.0)	(18.0–64.0)	(23.0–59.0)	(20.0–49.0)	(14.0–54.0)
Interval between 1st and 2nd dose (days)	N/A						
Median		23.0	70.0	28.0	70.0	21.0	21.0
[IQR]		[21.0–26.0]	[70.0–70.0]	[27.0–28.0]	[70.0–70.0]	[21.0–21.0]	[21.0–24.0]
(min–max)		(21.0–28.0)	(68.0–71.0)	(27.0–28.0)	(70.0–71.0)	(21.0–26.0)	(21.0–31.0)
Interval between the last dose and blood collection (days)	N/A						
Median		30.0	30.0	30.0	32.0	29.0	34.0
[IQR]		[27.0–32.0]	[26.0–31.0]	[30.0–32.0]	[30.0–32.0]	[29.0–35.0]	[31.0–35.0]
(min–max)		(27.0–33.0)	(21.0–35.0)	(30.0–33.0)	(30.0–33.0)	(25.0–40.0)	(22.0–35.0)

**Table 2 vaccines-11-01117-t002:** Characteristics of the participants who received three doses of COVID-19 vaccine.

Characteristics	SV/SV/SP	SV/SV/AZ	SV/SV/AZ (ID)	SV/SV/PF	SV/SV/PF (ID)
Number	29	76	31	56	30
Sex, Female no. (%)	16 (55.2)	52 (68.4)	24 (77.4)	33 (58.9)	20 (66.7)
Age years, mean	42.2	41.4	44.0	39.0	39.8
[SD]	[8.7]	[10.3]	[12.4]	[9.6]	[16.9]
(min–max)	(26.0–64.0)	(20.0–65.0)	(19.0–70.0)	(19.0–58.0)	(22.0–81.0)
Interval between 1st and 2nd dose (days)					
Median	21.0	21.0	21.0	27.0	23.0
[IQR]	[21.0–23.5]	[21.0–27.5]	[19.0–27.0]	[23.3–28.0]	[22.0–27.3]
(min–max)	(18.0–30.0)	(14.0–37.0)	(16.0–32.0)	(18.0–30.0)	(20.0–90.0)
Interval between 2nd and 3rd dose (days)					
Median	168.0	75.5	115.0	117.5	106.0
[IQR]	[164.0–172.5]	[60.3–140.8]	[103.0–117.0]	[76.0–142.5]	[66.0–107.5]
(min–max)	(115.0–188.0)	(45.0–191.0)	(43.0–139.0)	(24.0–164.0)	(46.0–135.0)
Interval between the last dose and blood collection (days)					
Median	28.0	28.0	30.0	28.0	30.0
[IQR]	[28.0–30.0]	[27.0–28.8]	[29.0–31.0]	[28.0–29.0]	[30.0–30.0]
(min–max)	(21.0–35.0)	(21.0–35.0)	(24.0–35.0)	(27.0–34.0)	(30.0–31.0)

**Table 3 vaccines-11-01117-t003:** Characteristics of the participants who received four doses of COVID-19 vaccine.

Characteristics	AZ/AZ/PF/PF	AZ/AZ/MO/MO½
Number	30	30
Sex, Female no. (%)	18 (60.0)	18 (60.0)
Age years, mean [SD]	52.0 [10.5]	52.7 [11.4]
(min–max)	(31.0–73.0)	(26.0–73.0)
Interval between 1st and 2nd dose (days)		
Median	81.0	81.0
[IQR]	[75.0–84.0]	[70.0–84.0]
(min-max)	(30.0–92.0)	(22.0–84.0)
Interval between 2nd and 3rd dose (days)		
Median	124.0	143.0
[IQR]	[117.0–148.5]	[111.0–170.0]
(min–max)	(86.0–204.0)	(84.0–201.0)
Interval between 3rd and 4th dose (days)		
Median	190.5	209.0
[IQR]	[144.5–229.0]	[186.8–224.0]
(min–max)	(94.0–248.0)	(136.0–250.0)
Interval between the last dose and blood collection (days)		
Median	28.0	31.0
[IQR]	[28.0–28.0]	[28.0–32.0]
(min–max)	(28.0–32.0)	(28.0–34.0)

## Data Availability

The datasets generated and analyzed during the current study are available from the corresponding author upon reasonable request.

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
