# Peer review of "Evaluation of Anti-S1 IgA Response to Different COVID-19 Vaccination Regimens"

_vaccines, 2023, doi:10.3390/vaccines11061117_

Round 1
Reviewer 1 Report
Evaluation of anti-S1 IgA response to different COVID-19 vaccination regimen
Authors evaluated the level of anti-S1 IgA in the serum of participants immunized with different COVID-19 vaccination regimen.
It is a work of interest, relevant to clinical practice.
Well written and clear in its objectives.
There are, however, some points that require attention and could be improved:
- The vaccination schedules should be presented in a table; I know that they are in the supplementary material with an indication of the amounts, but for those who are reading the manuscript, it would be important to have the schedule in an easier way to visualize, it does not need quantities, but which are the vaccines administered in each group, it can be as a table or with a drawing;
- When authors say that "We found that the routes and doses of vaccine immunization for either viral-vector or mRNA vaccines were not significantly different in the level of anti-S1 IgA. This result differed from that of a previous study by Intapiboon et al. ", they don't discuss this statement and they should, they must develop the subject, give hypotheses for the differences;
- When authors indicate the limitations of the study, the first two I think they will not be able to solve them at the moment, but as for evaluating the neutralization, I think they could do it and it would be an asset to the work
Reviewer 2 Report
The paper is well constructed and relevant since very few papers have studied IgA responses after COVID-19 vaccination. The cohort is also interesting for the variety of vaccines and regimens used. However, there are still a few issues that should be clarified.
1- The authors excluded from the study people who had COVID-19. It should be first be specified that only symptomatic COVID-19 could be excluded clinically. Moreover, how to be sure that the prevalence of asymptomatic infections is not a bias in this study? The different serum collection periods are in fact not specified. These periods may correspond to different waves of COVID-19 and thus to different frequencies of asymptomatic infections. The author should clarify this point.
2- The authors should explain on what basis people were selected to have different regimens
3- The authors should also explain on what basis the 10µg doses were chosen to the ID route
4- How the authors selected the subgroup of people for the follow-up of 4 months (we notice that there were 23 in each group)? The characteristics of these subgroups should be presented to be sure that there was no bias. Data comparing the IgA levels obtained by the different regimen at day 120 are also lacking.
Author Response
"Please see the attachment."

Round 2
Reviewer 1 Report
Evaluation of anti-S1 IgA response to different COVID-19 vaccination regimens
I believe that the issues raised have been taken into account, I understand the limitations to carrying out the neutralization tests. Maybe in the future you can consider a collaboration, it would improve the work without a doubt
Reviewer 2 Report
The manuscript is acceptable in this form